# CktGNN: Circuit Graph Neural Network for Electronic Design Automation

**Zehao Dong[1,†] Weidong Cao[2,†] Muhan Zhang[3] Dacheng Tao[4] Yixin Chen[1,‡] Xuan Zhang [2,‡] \***
{zehao.dong,weidong.cao,xuan.zhang}@wustl.edu
chen@cse.wustl.edu & muhan@pku.edu.cn & dacheng.tao@sydney.edu.au

## Abstract

The electronic design automation of analog circuits has been a longstanding challenge in the integrated circuit field due to the huge design space and complex design trade-offs among circuit specifications. In the past decades, intensive research efforts have mostly been paid to automate the transistor sizing with a given circuit topology. By recognizing the graph nature of circuits, this paper presents a Circuit Graph Neural Network (CktGNN) that simultaneously automates the circuit topology generation and device sizing based on the encoder-dependent optimization subroutines. Particularly, CktGNN encodes circuit graphs using a two-level GNN framework (of nested GNN) where circuits are represented as combinations of subgraphs in a known subgraph basis. In this way, it significantly improves design efficiency by reducing the number of subgraphs to perform message passing. Nonetheless, another critical roadblock to advancing learning-assisted circuit design automation is a lack of public benchmarks to perform canonical assessment and reproducible research. To tackle the challenge, we introduce Open Circuit Benchmark (OCB), an open-sourced dataset that contains 10K distinct operational amplifiers with carefully-extracted circuit specifications. OCB is also equipped with communicative circuit generation and evaluation capabilities such that it can help to generalize CktGNN to design various analog circuits by producing corresponding datasets. Experiments on OCB show the extraordinary advantages of CktGNN through representation-based optimization frameworks over other recent powerful GNN baselines and human experts' manual designs. Our work paves the way toward a learning-based open-sourced design automation for analog circuits. Our source code is available at `https://github.com/zehao-dong/CktGNN`.

## 1 Introduction

Graphs are ubiquitous to model relational data across disciplines (Gilmer et al., 2017; Duvenaud et al., 2015; Dong et al., 2021). Graph neural networks (GNNs) (Kipf & Welling, 2016; Xu et al., 2019; Velickovic et al., 2018; You et al., 2018; Scarselli et al., 2008) have been the de facto standard for representation learning over graph-structured data due to the superior expressiveness and flexibility. In contrast to heuristics using hand-crafted node features (Kriege et al., 2020) and non-parameterized graph kernels (Vishwanathan et al., 2010; Shervashidze et al., 2009; Borgwardt & Kriegel, 2005), GNNs incorporate both graph topologies and node features to produce the node/graph-level embeddings by leveraging inductive bias in graphs, which have been extensively used for node/graph classification (Hamilton et al., 2017; Zhang et al., 2018), graph decoding (Dong et al., 2022; Li et al., 2018), link prediction (Zhang & Chen, 2018), and etc. Recent successes in GNNs have boosted the requirement for benchmarks to properly evaluate and compare the performance of different GNN architectures. Numerous efforts have been made to produce benchmarks of various graph-structured data. Open Graph Benchmark (OGB) (Hu et al., 2020) introduces a collection of realistic and diverse graph datasets for real-world applications including molecular networks, citation networks,

---

*[1] Department of Computer Science & Engineering, Washington University in St. Louis. [2] Department of Electrical & Systems Engineering, Washington University in St. Louis. [3] Institute for Artificial Intelligence, Peking University. [4] School of Computer Science, The University of Sydney. [†] Equal contribution. [‡] Corresponding authors.

source code networks, user-product networks, etc. NAS-Bench-101 (Ying et al., 2019) and NAS-Bench-301 (Zela et al., 2022) create directed acyclic graph datasets for surrogate neural architecture search (Elsken et al., 2019; Wen et al., 2020). These benchmarks efficiently facilitate substantial and reproducible research, thereby advancing the study of graph representation learning.

Analog circuits, an important type of integrated circuit (IC), are another essential graph modality (directed acyclic graphs, i.e., DAGs). However, since the advent of ICs, labor-intensive manual efforts dominate the analog circuit design process, which is quite time-consuming and cost-ineffective. This problem is further exacerbated by continuous technology scaling where the feature size of transistor devices keeps shrinking and invalidates designs built with older technology. Automated analog circuit design frameworks are thus highly in demand. Dominant representation-based approaches (Liu et al., 2021; Wang et al., 2020; Cao et al., 2022a;b; Zhang et al., 2019a) have recently been developed for analog circuit design automation. Specifically, they optimize device parameters to fulfill desired circuit specifications with a given circuit topology. Typically, GNNs are applied to encode nodes' embeddings from circuit device features based on the fixed topology, where black-box optimization techniques such as reinforcement learning (Zoph & Le, 2016) and Bayesian Optimization (Kandasamy et al., 2018) are used to optimize parameterized networks for automated searching of device parameters. While these methods promisingly outperform traditional heuristics (Liu et al., 2017) in node feature sizing (i.e., device sizing), they are not targeting the circuit topology optimization/generation, which, however, constitutes the most critical and challenging task in analog circuit design.

In analogy to neural architecture search (NAS), we propose to encode analog circuits into continuous vectorial space to optimize both the topology and node features. Due to the DAG essence of analog circuits, recent DAG encoders for computation graph optimization tasks are applicable to circuit encoding. However, GRU-based DAG encoders (D-VAE (Zhang et al., 2019b) and DAGNN (Thost & Chen, 2021)) use shallow layers to encode computation defined by DAGs, which is insufficient to capture contextualized information in circuits. Transformer-based DAG encoder (Dong et al., 2022), however, encodes DAG structures instead of computations. Consequently, we introduce Circuit Graph neural Network (CktGNN) to address the above issues. Particularly, CktGNN follows the nested GNN (NGNN) framework (Zhang & Li, 2021), which represents a graph with rooted subgraphs around nodes and implements message passing between nodes with each node representation encoding the subgraph around it. The core difference is that CktGNN does not extract subgraphs around each node. Instead, a subgraph basis is formulated in advance, and **each circuit is modeled as a DAG $G$ where each node represents a subgraph in the basis**. Then CktGNN uses two-level GNNs to encode a circuit: the inner GNNs independently learn the representation of each subgraph as node embedding, and the outer GNN further performs directed message passing with learned node embeddings to learn a representation for the entire graph. The inner GNNs enable CktGNN to stack multiple message passing iterations to increase the expressiveness and parallelizability, while the outer directed message passing operation empowers CktGNN to encode computation of circuits (i.e. circuit performance).

Nonetheless, another critical barrier to advancing automated circuit design is the lack of public benchmarks for sound empirical evaluations. Researches in the area are hard to be reproduced due to the non-unique simulation processes on different circuit simulators and different search space design. To ameliorate the issue, we introduce Open Circuit Benchmark (OCB), the first open graph dataset for optimizing both analog circuit topologies and device parameters, which is a good supplement to the growing open-source research in the electronic design automation (EDA) community for IC (Chai et al., 2022; Hakhamaneshi et al., 2022). OCB contains 10K distinct operational amplifiers (circuits) whose topologies are modeled as graphs and performance metrics are carefully extracted from circuit simulators. Therefore, the EDA research can be conducted via querying OCB without notoriously tedious circuit reconstructions and simulation processes on the simulator. In addition, we will open-source codes of the communicative circuit generation and evaluation processes to facilitate further research by producing datasets with arbitrary sizes and various analog circuits. The OCB dataset is also going to be uploaded to OGB to augment the graph machine learning research.

The key contributions in this paper are: 1) we propose a novel two-level GNN, CktGNN, to encode circuits with deep contextualized information, and show that our GNN framework with a pre-designed subgraph basis can effectively increase the expressiveness and reduce the design space of a very challenging problem–circuit topology generation; 2) we introduce the first circuit benchmark dataset OCB with open-source codes, which can serve as an indispensable tool to advance research in EDA; 3) experimental results on OCB show that CktGNN not only outperforms competitive GNN baselines but also produces high-competitive operational amplifiers compared to human experts' designs.

## 2 RELATED WORKS

### 2.1 GRAPH NEURAL NETWORKS

**GNNs for DAGs** Directed acyclic graphs (DAGs) are another ubiquitous graph modality in the real world. Instead of implementing message passing across all nodes simultaneously, DAG GNNs (encoders) such as D-VAE (Zhang et al., 2019b) and DAGNN (Thost & Chen, 2021) sequentially encode nodes following the topological order. Message passing order thus respects the computation dependency defined by DAGs. Similarly, S-VAE (Bowman et al., 2016) represents DAGs as sequences of node strings of the node type and adjacency vector of each node and then applies a GRU-based RNN to the topologically sorted sequence to learn the DAG representation. To improve the encoding efficiency, PACE (Dong et al., 2022) encodes the node orders in the positional encoding and processes nodes simultaneously under a Transformer (Vaswani et al., 2017) architecture.

### 2.2 AUTOMATED ANALOG CIRCUIT DESIGN

**Design Automation Methods for Device Sizing** Intensive research efforts have been paid in the past decades to automate the analog circuit design at the pre-layout level, i.e., finding the optimal device parameters to achieve the desired circuit specifications. Early explorations focus on optimization-based methods, including Bayesian Optimization (Lyu et al., 2018), Geometric Programming (Colleran et al., 2003), and Genetic Algorithms (Liu et al., 2009). Recently, learning-based methods such as supervised learning methods (Zhang et al., 2019a) and reinforcement learning methods (Wang et al., 2020; Li et al., 2021; Cao et al., 2022a;b) have emerged as promising alternatives. Supervised learning methods aim to learn the underlying static mapping relationship between the device parameters and circuit specifications. Reinforcement learning methods, on the other hand, endeavor to find a dynamic programming policy to update device parameters in an action space according to the observations from the state space of the given circuit. Despite their great promise, all these prior arts have been limited to optimizing the device parameters with a given analog circuit topology. There are only a few efforts (e.g., Genetic Algorithms (Das & Vemuri, 2007)) to tackle another very challenging yet more important problem, i.e, circuit topology synthesis. These works leverage genetic operations such as crossover and mutation to randomly generate circuit topologies and do not sufficiently incorporate practical constraints from feasible circuit topologies into the generation process. Therefore, most of the generated topologies are often non-functional and ill-posed. Conventionally, a newly useful analog circuit topology is manually invented by human experts who have rich domain knowledge within several weeks or months. Our work focuses on efficiently and accurately automating circuit topology generation, based on which the device parameters for the circuit topology are further optimized.

**Graph Learning for Analog Circuit Design Automation** With the increasing popularity of GNNs in various domains, researchers have recently applied GNNs to model circuit structures as a circuit topology resembles a graph very much. Given a circuit structure, the devices in the circuit can be treated as graph vertices, and the electrical connections between devices can be abstracted as edges between vertices. Inspired by this homogeneity between the circuit topology and graph, several prior arts have explored GNNs to automate the device sizing for analog circuits. A supervised learning method (Zhang et al., 2019a) is applied to learn the geometric parameters of passive devices with a customized circuit-topology-based GNN. And reinforcement learning-based methods (Wang et al., 2020; Cao et al., 2022b) propose circuit-topology-based policy networks to search for optimal device parameters to fulfill desired circuit specifications. Distinctive from these prior arts, our work harnesses a two-level GNN encoder to simultaneously optimize circuit topologies and device features.

## 3 CIRCUIT GRAPH NEURAL NETWORK

In this section, we introduce the proposed CktGNN model constructed upon a two-level GNN framework with a subgraph basis to reduce the topology search space for the downstream optimization algorithm. We consider the graph-level learning task. Given a graph $\mathcal{G} = (V, E)$, where $V = \{1, 2, ..., n\}$ is the node set with $|V| = n$ and $E \in V \times V$ is the edge set. For each node $i$ in a graph $\mathcal{G}$, we let $\mathcal{N}(v) = \{u \in V | (u, v) \in E\}$ denote the set of neighboring nodes of $v$.

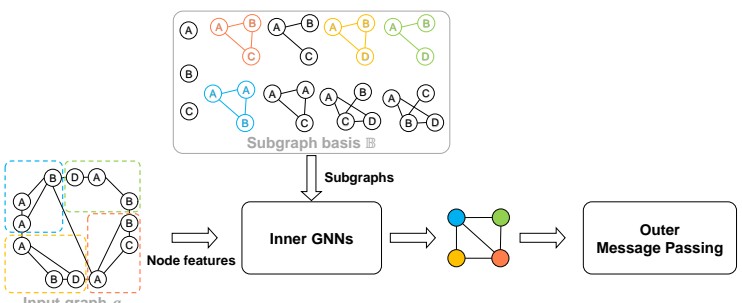

Figure 1: Illustration of the two-level GNN framework with a pre-designed subgraph basis. It first represents input graph $g$ as a combination of subgraphs in the subgraph basis $\mathbb{B}$ and then learns representations of subgraphs with inner GNNs. The subgraph representations are used as input features to the outer message passing operation, where the message passing can be directed/undirected.

## 3.1  TWO-LEVEL GNN FRAMEWORK WITH A SUBGRAPH BASIS

Most undirected GNNs follow the message passing framework that iteratively updates the nodes' representation by propagating information from the neighborhood into the center node. Let $h_v^t$ denote the representation of $v$ at time stamp $t$, the message passing framework is given by:

$$a_v^{t+1} = \mathcal{A}(\{h_u^t|(u,v) \in E\}) \quad h_v^{t+1} = \mathcal{U}(h_v^t, a_v^{t+1}) \tag{1}$$

Here, $\mathcal{A}$ is an aggregation function on the multiset of representations of nodes in $\mathcal{N}(v)$, and $\mathcal{U}$ is an update function. Given an undirected graph $G$, GNNs perform the message passing over all nodes simultaneously. For a DAG $G$, the message passing progresses following the dependency of nodes in DAG $G$. That is, a node $v$'s representation is not updated until all of its predecessors are processed.

It has been shown that the message passing scheme mimics the 1-dimensional Weisfeiler-Lehman (1-WL) algorithm (Leman & Weisfeiler, 1968). Then, the learned node representation encodes a rooted subtree around each node. And GNNs exploit the homophily as a strong inductive bias in graph learning tasks, where graphs with common substructures will have similar predictions. However, encoding rooted subtrees limits the representation ability, and the expressive power of GNNs is upper-bounded by the 1-WL test (Xu et al., 2019). For instance, message passing GNNs fail to differentiate d-regular graphs (Chen et al., 2019; Murphy et al., 2019).

To improve the expressive ability, Zhang & Li (2021) introduces a two-level GNN framework, NGNN, that encodes the general local rooted subgraph around each node instead of a subtree. Concretely, given a graph $G$, $h$-hop rooted subgraph $g_v^h$ around each node $v$ is extracted. Then, inner GNNs are independently applied to these subgraphs $\{g_v^h|v \in G\}$ and the learned graph representation of $g_v^h$ is used as the input embedding of node $v$ to the outer GNN. After that, the outer GNN applies graph pooling to get a graph representation. The NGNN framework is strictly more powerful than 1-WL and can distinguish almost all $d$-regular graphs. However, the two-level GNN framework can not be applied to DAG encoders (GNNs).

Hence, we introduce a two-level GNN framework with an (ordered) subgraph basis, to restrict subgraphs for inner message passing in the given subgraph basis and apply it to the circuit (DAG) encoding problems in order to reduce the topology search space and increase the expressive ability.

**Definition 3.1** *(Ordered subgraph basis) An ordered subgraph basis $\mathbb{B} = \{g_1, g_2, ..., g_K\}$ is a set of subgraphs with a total order o. For $\forall g_1,\ g_2 \in \mathbb{B}$, $g_1 < g_2$ if and only if $o(g_1) < o(g_2)$.*

Figure 1 illustrates the two-level GNN framework. Given a graph $G$ and an ordered subgraph basis $\mathbb{B}$, the rules to extract subgraphs for inner GNNs to learn representations are as follows: 1) For node $v \in G$, suppose it belongs to multiple subgraphs $g_1^v, ...g_m^v \in \mathbb{B}$, then the selected subgraph to perform (inner) message passing is $g_h^v = argmax_{i=1,2,...,m}o(g_i^v)$. 2) If connected nodes $v$ and $u$ select the same subgraph $g_h \in \mathbb{B}$, we merge $v$ and $u$ and use the representation of subgraph $g_h$ as the feature of the merged node when performing the outer message passing. We show that two-level GNNs can be more powerful than 1-WL in Appendix B.

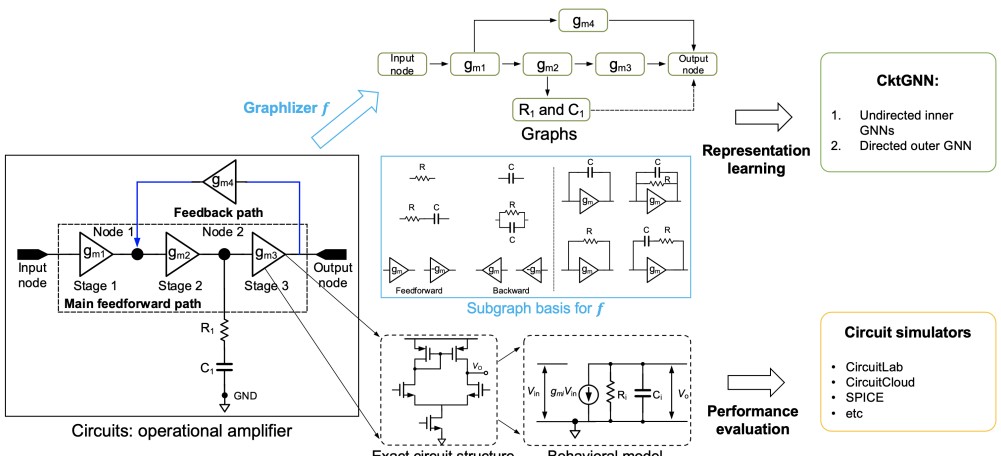

Figure 2: Illustration of the overall framework. The performance evaluation of circuits is implemented on the circuit simulator. In the representation learning process, we formulate a subgraph basis for operational amplifiers to implement the CktGNN model.

## 3.2 THE CKTGNN MODEL

Next, we introduce the CktGNN model for the circuit (i.e. DAGs) automation problem which requires optimizing the circuit topology and node features at the same time. Given a circuit $G = (V, E)$, each node $v \in V$ has a node type $x_t$ and node features $x_s$, where $x_t$ denotes the device type in a circuit (i.e., resistor, capacitor, and etc.), and $x_s$ are the categorical or continuous features of the corresponding device.

Due to the similarity between the circuit automation problem and neural architecture search (NAS), potential solutions naturally come from advancements in neural architecture search, where two gradient-based frameworks have achieved impressive results in DAG architecture learning: 1) The VAE framework (Dong et al., 2022; Zhang et al., 2019b) uses a GRU-based (or Transformer-based) encoder to map DAGs into vectorial space and then trains the model with a VAE loss. 2) NAS-subroutine-based framework develops DAG encoders and implements encoding-dependent subroutines such as perturb-architecture subroutines (Real et al., 2019; White et al., 2020) and train-predictor-model subroutines (Wen et al., 2020; Shi et al., 2019). As the NAS-subroutine-based framework usually requires a relatively large-size training dataset, yet the large-scale performance simulation of circuits can be time-consuming, we resort to the VAE framework.

Due to the complexity of circuit (DAG) structures and the huge size of the circuit design space (see Appendix A), the circuit automation problem is typically a highly non-convex, challenging optimization problem. Thus, the GRU-based encoders that use shallow layers to encode complex DAG architectures are not sufficient to capture complicated contextualized information of node features and topologies.

To address these limitations, we introduce the CktGNN model. Figure 2 illustrates the architecture of CktGNN. The key idea is to decompose the input circuit as combinations of non-overlapping subgraphs in the subgraph basis $\mathbb{B}$. After the graph transformation process (i.e. graphlizer $f$), each node in the transformed DAG $G' = (V', E')$ represents a subgraph in the input graph (circuit). Then, the representation of a circuit is learned in the two-level GNN framework.

Each node $v' \in V'$ in the transformed DAG $G'$ corresponds to a subgraph $g_{v'}$ in the input graph. CktGNN treats $g_{v'}$ as an undirected graph and uses inner GNNs to learn the subgraph representation $h_{v'}$, and each inner GNN consists of multiple undirected message passing layers followed by a graph pooling layer to summarize a subgraph representation. Such a technique enables inner GNNs to capture the contextualized information within each subgraph, thereby increasing the representation ability. In addition, undirected message passing also provides better parallelizability than directed message passing, which increases encoding efficiency.

In the outer level GNN, CktGNN performs directed message passing where the aggregation function $\mathcal{A}$ uses a gated summation and the update function $\mathcal{U}$ uses a gated recurrent unit (GRU):

$$a_{v'} = \mathcal{A}(\{z_{u'}|u^{'} \in \mathcal{N}(v^{'})\}) = \sum_{u' \in \mathcal{N}(v')} g(z_{u'}) \otimes m(z_{u'}), \quad (2)$$

$$z_{v'} = \mathcal{U}(concat(x^{'}_{v'}, h_{v'}), a_{v'}). \quad (3)$$

Here, $z_{v'}$ is the hidden representation of node $v^{'}$ in the transformed DAG $G^{'}$, $m$ is a mapping network and $g$ is a gating network. In the GRU, $x^{'}_{v'}$ is the one-hot encoding of the subgraph type, and $h_{v'}$ is the corresponding subgraph representation learned by inner GNNs. The outer level GNN processes nodes in transformed DAG $G^{'}$ following the topological order, and uses the hidden representation of the output node as the graph representation.

## 3.3  DISCUSSIONS

**The Injectivity of Graph Transformation Process** Since CktGNN is constructed upon a graphlizer $f : G \rightarrow G^{'}$ that converts input circuits $G$ to DAGs $G^{'}$ whose nodes $v^{'}$ represent non-overlapping subgraphs in $G$, it's worth discussing the injectivity of $f$. If $f(G)$ is not unique, CktGNN will map the same input circuit (DAG $G$) to different transformed $G^{'}$, thereby learning different representations for the same input $G$.

**Theorem 3.2** *Let subgraph basis $\mathbb{B}$ contain every subgraph of size 1. There exists an injective graph transformation $f$, if each subgraph $g_{v'} \in \mathbb{B}$ has only one node to be the head (tail) of a directed edge whose tail (head) is outside the subgraph $g_{v'}$.*

We prove Theorem 3.2 in Appendix C. The theorem implies that $f$ exists when subgraph basis $\mathbb{B}$ contains every subgraph of size 1 and characterizes conditions when the injectivity concern is satisfied. The proof shows that an injective $f$ can be constructed based on the order function $o$ over the basis $\mathbb{B}$. The condition in the Theorem holds in various real-world applications including circuit automation and neural architecture search. For instance, Figure 2 presents an ordered subgraph basis that satisfies the condition for the general operational amplifiers (circuits), and we introduce operational amplifiers and the corresponding subgraph basis in Appendix A.

**Comparison to Related Works.** The proposed CktGNN model extends the NGNN technique (i.e., two-level GNN framework) (Zhang & Li, 2021) to the directed message passing framework for DAG encoding problem. In contrast to NGNN which extracts a rooted subgraph structure around each node within which inner GNNs perform message passing to learn the subgraph representation, CktGNN only uses inner GNNs to learn representations of non-overlapping subgraphs in a known ordered subgraph basis $\mathbb{B}$. Two main advantages come from the framework over GRU-based DAG encoders (i.e. D-VAE and DAGNN). 1) The topology within subgraphs is automatically embedded once the subgraph type is encoded in CktGNN, which significantly reduces the size of the topology search space. 2) The inner GNNs in CktGNN help to capture the contextualized information within each subgraph, while GNNs in GRU-based DAG encoders are usually too shallow to provide sufficient representative ability.

## 4  OPEN CIRCUIT BENCHMARK

We take operational amplifiers (Op-Amps) to build our open circuit benchmark as they are not only one of the most difficult analog circuits to design in the world, but also are the common benchmarks used by prior arts (Wang et al., 2020; Li et al., 2021; Cao et al., 2022b) to evaluate the performance of the proposed methods. Our benchmark equips with communicative circuit generation and evaluation capabilities such that it can also be applied to incorporate a broad range of analog circuits for the evaluations of various design automation methods. To enable such great capabilities, two notable features are proposed and introduced below.

**Converting Circuits into Graphs** We leverage an acyclic graph mapping method by abstracting a complex circuit structure to several simple low-level functional sub-structures, which is similar to the standard synthesis of modern digital circuits. Such a conversion method is thus scalable to large-scale analog circuits with thousands of devices while effectively avoiding cycles in graph converting. To

Table 1: Predictive performance and topology reconstruction accuracy.

| | Gain | | BW | | PM | | FoM | | Recon |
|---|---|---|---|---|---|---|---|---|---|
| Evaluation Metric | RMSE ↓ | Pearson's r ↑ | RMSE ↓ | Pearson's r ↑ | RMSE ↓ | Pearson's r ↑ | RMSE ↓ | Pearson's r ↑ | Acc ↑ |
| **CktGNN** | **0.607 ± 0.003** | **0.791 ± 0.002** | **0.873 ± 0.003** | **0.479 ± 0.001** | 0.973 ± 0.002 | 0.217 ± 0.001 | **0.854 ± 0.003** | **0.491 ± 0.002** | **0.397** |
| PACE | 0.644 ± 0.003 | 0.762 ± 0.002 | 0.896 ± 0.003 | 0.442 ± 0.001 | 0.970 ± 0.003 | 0.226 ± 0.001 | 0.889 ± 0.003 | 0.423 ± 0.001 | 0.306 |
| DAGNN | 0.695 ± 0.002 | 0.707 ± 0.001 | 0.881 ± 0.002 | 0.453 ± 0.001 | 0.969 ± 0.003 | 0.231 ± 0.002 | 0.877 ± 0.003 | 0.442 ± 0.001 | 0.289 |
| D-VAE | 0.681 ± 0.003 | 0.739 ± 0.001 | 0.914 ± 0.002 | 0.394 ± 0.001 | **0.956 ± 0.003** | **0.301 ± 0.002** | 0.897 ± 0.003 | 0.374 ± 0.001 | 0.271 |
| GCN | 0.976± 0.003 | 0.140 ± 0.002 | 0.970 ± 0.003 | 0.236 ± 0.001 | 0.993 ± 0.002 | 0.171 ± 0.001 | 0.974 ± 0.003 | 0.217 ± 0.001 | 0.058 |
| GIN | 0.890 ± 0.003 | 0.352 ± 0.001 | 0.926 ± 0.002 | 0.251 ± 0.001 | 0.985 ± 0.004 | 0.187 ± 0.002 | 0.910 ± 0.003 | 0.284 ± 0.001 | 0.051 |
| NGNN | 0.882 ± 0.004 | 0.433 ± 0.002 | 0.933 ± 0.003 | 0.247 ± 0.001 | 0.984 ± 0.004 | 0.196 ± 0.002 | 0.926 ± 0.002 | 0.267 ± 0.001 | 0.068 |
| Pathformer | 0.816 ± 0.003 | 0.529 ± 0.001 | 0.895 ± 0.002 | 0.410 ± 0.001 | 0.967 ± 0.002 | 0.297 ± 0.001 | 0.887 ± 0.002 | 0.391 ± 0.001 | 0.081 |

illustrate this general mapping idea, we take the Op-Amps in our benchmark as an example (see the left upper corner in Figure 2). For an $N$-stage Op-Amp ($N = 2, 3$), it consists of $N$ single-stage Op-Amps in the main feedforward path (i.e., from the input direction to the output direction) and several feedback paths (i.e., vice versa) with different sub-circuit modules. We encode all single-stage Op-Amps and sub-circuit modules as functional sub-structures by using their behavioral models (Lu et al., 2021). In this way, each functional sub-structures can be significantly simplified without using its exact yet complex circuit structure. For instance, a single-stage Op-Amp with tens of transistors can be modeled as a voltage-controlled current source (VCCS, $g_m$) with a pair of parasitic capacitor $C$ and resistor $R$. Instead of using these functional sub-structures as graph vertices, we leverage them as our graph edges while the connection point (e.g., node 1 and node 2) between these sub-structures is taken as vertices. Meanwhile, we unify both feedforward and feedback directions as feedforward directions but distinguish them by adding a polarity (e.g., '+' for feedforward and '−' for feedback) on device parameters (e.g., $g_m+$ or $g_m-$). In this way, an arbitrary Op-Amp can be efficiently converted into an acyclic graph as shown in Figure 2. Inversely, a newly-generated circuit topology from graph sampling can be converted back into the corresponding Op-Amp by using a conversion-back process from the graph and functional sub-structures to real circuits.

**Interacting with Circuit Simulators** Another important feature of our circuit benchmark is that it can directly interact with a circuit simulator to evaluate the performance of a generated Op-Amp in real-time. Once a circuit topology is generated from our graph sampling, a tailored Python script can translate it into a circuit netlist. A circuit netlist is a standard hardware description of a circuit, based on which the circuit simulator can perform a simulation (evaluation) to extract the circuit specifications, e.g., gain, phase margin, and bandwidth for Op-Amps. This process can be inherently integrated into a Python environment as both open-sourced and commercial circuit simulators support command-line operations. We leverage this conversion-generation-simulation loop to generate $10,000$ different Op-Amps with detailed circuit specifications. Note that in our dataset, the topologies of Op-Amps are not always distinct from each other. Some Op-Amps have the same topology but with different device parameters. However, our benchmark is friendly to augment other analog circuits such as filters if corresponding circuit-graph mapping methods are built.

## 5 EXPERIMENTS

### 5.1 DATASET, BASELINES, AND TASKS

**Dataset** Our circuit dataset contains 10,000 operational amplifiers (circuits) obtained from the circuit generation process of OCB. Nodes in a circuit (graph) are sampled from $C$ (capacitor), $R$ (resistor), and single-stage Op-Amps with different polarities (positive or negative) and directions (feedforward or feedback). Node features are then determined based on the node type: resistor $R$ has a specification from $10^5$ to $10^7$ Ohm, capacitor $C$ has a specification from $10^{-14}$ to $10^{-12}$ F, and the single-stage Op-Amp has a specification (transconductance, $g_m$) from $10^{-4}$ to $10^{-2}$ S. For each circuit, the circuit simulator of OCB performs careful simulations to get the graph properties (circuit specifications): DC gain (Gain), bandwidth (BW), and phase margin (PM), which characterize the circuit performance from different perspectives. Then the Figure of Merit (FoM) which is an indicator of the circuit's overall performance is computed from Gain, BW, and PM. Details are available in Appendix B.

**Baselines** We compare CktGNN with GNN baselines including: 1) widely adopted (undirected) GNNs: GCN (Kipf & Welling, 2016), GIN (Xu et al., 2019), NGNN (Zhang & Li, 2021) and

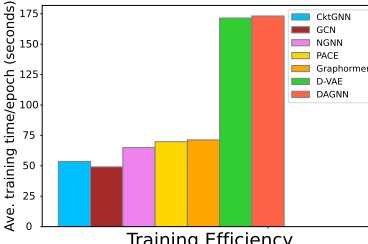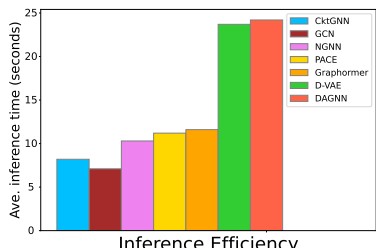

Figure 3: Comparison of training/inference efficiency.

Graphormer (Ying et al., 2021); 2) dominant DAG encoders (directed GNNs): D-VAE (Zhang et al., 2019b), DAGNN (Thost & Chen, 2021) and PACE (Dong et al., 2022). Baseline settings are provided in Appendix D.

**Tasks** We compare CktGNN against baselines on the following three tasks to evaluate the expressiveness, efficiency, and real-world impact: (1) Predictive performance and topology reconstruction accuracy: We test how well the learned graph representation encodes information to predict the graph properties (i.e., Gain, BW, PM, FoM) and reconstruct the input circuit topology. (2) Circuit encoding efficiency: This task compares the training/inference time to characterize the efficiency of the circuit (DAG) encoders. (3) Effectiveness in real-world electronic circuit design automation: For the purpose of circuit generation, we test the proportion that the decoder in the VAE architecture can generate valid DAGs, circuits, and new circuits that are never seen in the training set. Furthermore, for the purpose of automated circuit design, we also perform Bayesian Optimization and compare the overall performance (i.e. FoM) of the detected circuit topology and specifications.

## 5.2 Predictive Performance and Topology Reconstruction Accuracy

In the experiment, we test the expressive ability of circuit encoders by evaluating the predictivity of the learned graph representation. As the encoder with more expressive ability can better distinguish circuits (graphs) with different topologies and specifications, circuits with similar properties (Gain, BW, PM, or FoM) can be mapped to close representations. Following experimental settings of D-VAE (so as DAGNN and PACE), we train sparse Gaussian Process (SGP) regression models (Snelson & Ghahramani, 2005) to predict circuit properties based on learned representations. In addition, we also characterize the complexity of topology space to encode by measuring the reconstruction accuracy of the circuit topology.

We show our results in Table 1. Experimental results illustrate that CktGNN consistently achieves state-of-art performance in predicting circuit properties. In analogy to D-VAE, we find CktGNN encodes computation (i.e., circuit performance) instead of graph structures when the map between subgraph structure and their defined computation is injective for subgraphs in the basis $\mathbb{B}$. Compared to DAG encoders D-VAE and DAGNN that encodes circuit, inner GNNs in CktGNN enables more message passing iterations to better encode the complicated contextualized information in circuits. On the other hand, PACE, undirected GNNs, and graph Transformers inherently encode graph structures, hence, the latent graph representation space of CktGNN is smoother w.r.t the circuit performance. Furthermore, we also find that CktGNN significantly improves the topology reconstruction accuracy. Such an observation is consistent with the purpose of CktGNN (see Section 3.2) to reduce the topology search space.

## 5.3 Circuit Encoding Efficiency

Besides the representation learning ability, efficiency and scalability play critical roles in the model design due to the potential heavy traffic coming into the circuit design automation problem in the real world. In the experiment, we compare the average training time per epoch to validate the efficiency of the VAE framework that includes a circuit encoding process and a circuit generation process, while the encoding efficiency itself is tested using the average inference time. Figure 3 illustrates the results. Compared to GRU-based computation encoders (i.e., D-VAE and DAGNN), since CktGNN utilizes simultaneous message passing within each subgraph, it significantly reduces the training/inference

time. In addition, we also find that the extra computation time of CktGNN to fully parallelize encoders (i.e., undirected GNNs like GCN) is marginal.

## 5.4 EFFECTIVENESS IN REAL-WORLD ELECTRONIC CIRCUIT DESIGN

Next, we test the real-world performance of different circuit encoders from two aspects. 1) In a real-world automation process, the circuit generator (i.e., decoder in the VAE framework) is required to be effective to generate proper and novel circuits. Hence, we compare the proportion of valid DAGs, valid circuits, and novel circuits of different methods. 2) We also perform batch Bayesian Optimization (BO) with a batch size of 50 using the expected improvement heuristic (Jones et al., 1998) and compute the FoM of the detected circuits after 10 iterations.

Table 2: Effectiveness in real-world electronic circuit design.

| Methods | Valid DAGs (%) ↑ | Valid circuits (%) ↑ | Novel circuits (%) ↑ | BO (FoM) ↑ |
|---|---|---|---|---|
| **CktGNN** | **98.92** | **98.92** | 92.29 | **33.436447** |
| PACE | 83.12 | 75.52 | 97.14 | 33.274162 |
| DAGNN | 83.10 | 74.21 | 97.19 | 33.274162 |
| D-VAE | 82.12 | 73.93 | **97.15** | 32.377778 |
| GCN | 81.02 | 72.03 | 97.01 | 31.624473 |
| GIN | 80.92 | 73.17 | 96.88 | 31.624473 |
| NGNN | 82.17 | 73.22 | 95.29 | 32.282656 |
| Graphormer | 82.81 | 72.70 | 94.80 | 32.282656 |

Table 2 illustrates the results. We find that the proportion of valid circuits generated by the CktGNN-related decoder is significantly higher than other decoders. One potential reason is that CktGNN has an easier topology space to encode and the corresponding decoder can thus better learn the circuit generation rules. We also find that CktGNN has the best circuit design (generation) ability in the generation process. These observations are significant for real-world applications as the automation tools can perform fewer simulations to be cost-effective and time-efficient. In the end, we find that CktGNN-based VAE can find the best circuits with the highest FoM, and we visualize detected circuits in Appendix D.

## 6 CONCLUSION AND DISCUSSIONS

In this paper, we have presented CktGNN, a two-level GNN model with a pre-designed subgraph basis for the circuit (DAG) encoding problem. Inspired by previous VAE-based neural architecture search routines, we applied CktGNN based on a VAE framework for the challenging analog circuit design automation task. Experiments on the proposed open circuit benchmark (OCB) show that our automation tool can address this long-standing challenging problem in a predictivity-effective and time-efficient way. To the best of our knowledge, the proposed CktGNN-based automation framework pioneers the exploration of learning-based methods to simultaneously optimize the circuit topology and device parameters to achieve the best circuit performance. In addition, the proposed OCB is also the first open-source benchmark in the field of analog circuit topology generation and device parameter optimization. Last but not the least, both our method and benchmark can be generalized to other types of analog circuits with excellent scalability and compatibility.

With the increasing popularity of applying deep learning methods to design industrial digital circuits such as Google TPU (Mirhoseini et al., 2021) and Nvidia GPU (Khailany et al., 2020), the attention paid to analog circuit design automation will be unprecedented as analog circuits are a critical type of ICs to connect our physical analog world and modern digital information world. In a nutshell, we believe that deep learning (especially graph learning)-based analog circuit design automation is an important rising field, which is worthy of extensive explorations and interdisciplinary collaborations.

## 7 ACKNOWLEDGEMENT

This work is supported in part by NSF CCF #1942900 and NSF CBET 2225809.

## 8 REPRODUCIBILITY STATEMENT

The main theoretical contribution of our paper comes from Theorem 3.2, and the complete proof of the Theorem is available in Appendix C. Furthermore, our proposed circuit encoder, CktGNN, is constructed upon the two-level GNN framework with a pre-designed subgraph basis, hence we compare it with general two-level GNN in Section 3.2, and discuss the expressive ability in Appendix B. For our open circuit benchmark (OCB), we provide detailed instructions in Appendix A, and provide open-source code in supplementary material, including the circuit generation code and the simulation file for the circuit simulator. Our source code to implement experiments is also provided in the supplementary materials, and we will make it public on Github in the future.

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

# A    OPERATIONAL AMPLIFIERS (CIRCUITS) AND THE CORRESPONDING SUBGRAPH BASIS

In this work, we take one type of the most classical analog circuits, i.e., operational amplifiers (Op-Amps), as an example to present our graph learning-based automation framework. In this paper, the question we are interested in is that given a group of Op-Amp circuit specifications that are required by a particular application, how do we use a graph learning-based automation method to find a proper Op-Amp topology and the corresponding optimal device parameters to achieve the desired circuit specifications? However, our ultimate goal is to generalize this method to design various analog circuits and even radio-frequency and millimeter-wave circuits.

For an Op-Amp, it has many circuit specifications. We pick up four main specifications, i.e., DC gain (A), bandwidth (BW), phase margin (PM), and power consumption (P) in this project. All these specifications can be simulated by using circuit simulators or derived by using a simplified behavioral model of Op-Amps.

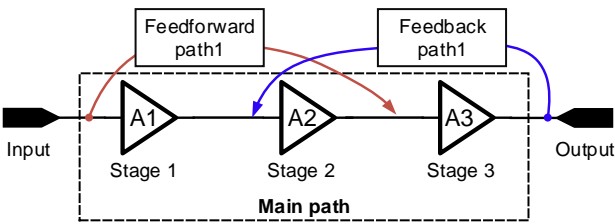

Figure 4: The conceptual illustration of a three-stage op-amp.

Figure 4 shows the conceptual illustration of a three-stage Op-Amp, which consists of the main path, several feed-forward paths (e.g., feed-forward path 1), and several feedback paths (e.g., feedback path1). In the main path, there are three single-stage Op-Amps, i.e., A1, A2, and A3. The total gain of the three-stage Op-Amp is the product of the gain of each single-stage Op-Amp. To make the Op-Amp more stable, the feed-forward and feedback paths are used to realize the compensation schemes. Each of these paths is implemented with electronic circuits, e.g., single-stage Op-Amps, resistor, and capacitor circuits. Note that, the forward and feedback paths are versatile. Figure. 1 just shows a very simple example. In addition, each single-stage Op-Amp could have many different topologies. For example, Figure 5 shows some single-stage Op-Amps with various topologies.

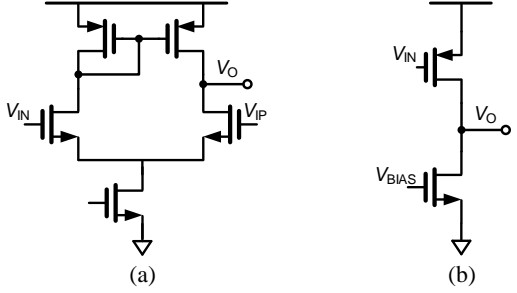

Figure 5: Different topologies of single-stage Op-Amps. (a) A differential Op-Amp. (b) A single-end Op-Amp.

Due to the different topologies of single-stage Op-Amps and various compensation schemes, the design space (e.g., topology space and device parameter space) of three-stage Op-Amps is thus very large. The manual process adopted by a human designer is very time-consuming. That is why graph learning-based methods come here. We can formulate the design of three-stage Op-Amps as a graph generation and optimization problem. In this way, we can automate the design process. Therefore, the first step is to model the circuit as a graph. Figure 6 shows the basic idea. We map the topology of a three-stage Op-Amp into a graph with five nodes: input node, node 1, node 2, output node, and

GND node. The edge connecting two nodes can be a single-stage Op-Amp (i.e., transconductance stage, $g_{mi}$), resistor (R) or capacitor (C), or a combination of them, which are discussed below.

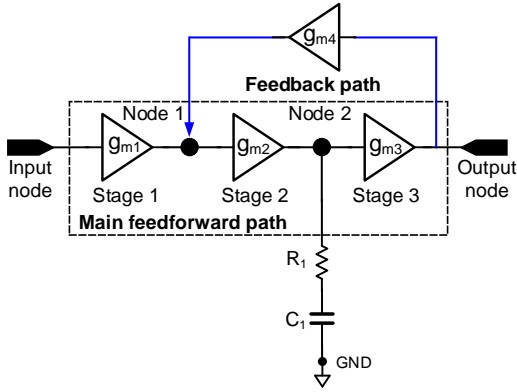

Figure 6: Graph mapping of a three-stage Op-Amp.

From the perspective of designing a three-stage Op-Amp, there are 24 types of (existing) connections between every two nodes in the circuit graph shown in Figure 6:

- Only single C or R connection (Figure 7(a), 2 types)
- C and R connected in parallel or serial (Figure 7(b), 2 types)
- A single-stage Op-Amp ($g_m$) with different polarities (positive, $+g_m$, or negative, $-g_m$) and directions (feedforward or feedback) (Figure 7(c), 4 types);
- A single-stage Op-Amp ($gm$) with C or R connected in parallel or serial (Figure 7(d), 16 types); note that here we only take the single-stage Op-Amp with feedforward direction and positive polarities as an example

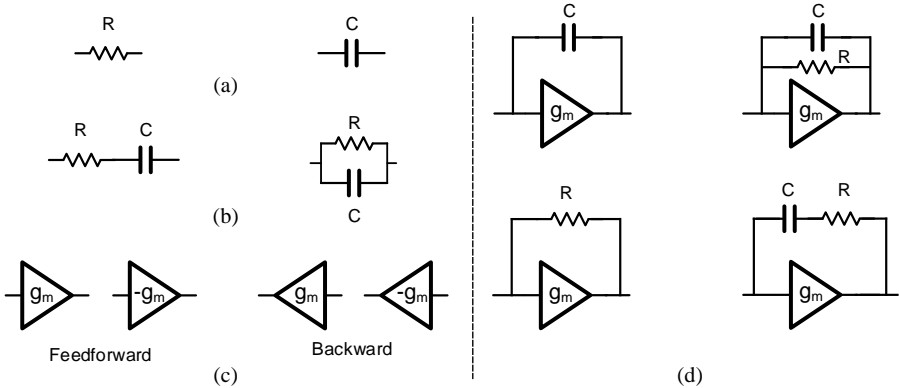

Figure 7: Subgraph basis for operational amplifiers.

To simplify the graph learning process, we convert the role of nodes and edges (except for the input node and output node) in Figure 6, then the Figure 7, which illustrates the edge types in the original graph, now presents a subgraph basis $\mathbb{B}$ for the operational amplifier.

Given the pre-designed subgraph basis $\mathbb{G}$, we also need to define a total order $o$ on subgraphs in $\mathbb{B}$ to guarantee that circuits are injectively represented as DAGs $G'$. Basically, for subgraphs of size 1, we set $o(g_m) > o(\text{R}) > o(\text{C})$, and $g_m$ with feedforward direction and positive polarity as a prior order than $g_m$ with backward direction and negative polarity. For subgraphs of size larger than 1, the order of subgraphs is firstly dependent on the number of nodes in the subgraph. Then, for subgraphs of the same size, the one with a parallel connection is prior to that with a series connection. In the end,

if the connection type is also the same, the order of subgraphs is determined by lexicographically sorting the order of nodes in subgraphs.

**Expressiveness and total order on nodes**     Various node orderings have been adopted in encoders for the DAG encoding problem, and node ordering is especially important for auto-regressive models. For instance, S-VAE, D-VAE and DAGNN use topological ordering, PACE and GRAN Liao et al. (2019) use canonical ordering, and graphRNN You et al. (2018) uses BFS ordering. The key of developing powerful DAG encoders is to make sure that node-order dependent encoding process can injectively map non-isomorphic DAGs (or different computation of DAGs) to different embeddings in the vectorial space, and experimental results in prior works (PACE, D-VAE, DAGNN) have confirmed the claim on general DAG encoding tasks, such as NAS (neural architecture search) and Bayesian network structure learning. In summary, PACE can injectively encode the structure of DAGs by linearizing DAGs as sequences according to the canonical ordering, while D-VAE and DAGNN can injectively encode computation of DAGs following an RNN-based directed message passing. Compared with these methods, S-VAE and GraphRNN-like encoder (GraphRNN is originally designed as a pure generative model, yet it can be extended to a BFS-order based DAG encoder following the idea of S-VAE, and details are available in the ablation study section in paper of PACE) fail to injectively encode structure nor computation of DAGs as both topological ordering and BFS ordering are not unique. As a result, PACE, D-VAE, and DAGNN significantly outperforms S-VAE and GraphRNN.

Following the above analysis, we find that the expressive ability of CktGNN is independent of the domain specific total ordering of nodes. Basically, total ordering of nodes only determines the order of subgraphs (i.e. $o$) in the subgraph basis $\mathbb{B}$ (plases see the last paragraph in Appendix A). When basis $\mathbb{B}$ satisfies the conditions in Theorem 3.2, there exists an injective graphlizer $f$, and $f$ can be formulated as Appendix C suggests. Hence, for **any given order** $o$, the input graph $G$ is injectively mapped to $f(G)$, based on which CktGNN performs the two-level GNNs. Then, if both the inner GNNs and outer directed message passing are injective, CktGNN can injectively map $f(G)$ to embeddings in the vectorial space, as injective functions follow the transitive property. Hence, for any order $o$ on subgraph basis $\mathbb{B}$, the overall encoding process can injectively map $G$ to embeddings.

Although the expressive ability is not affected by the node ordering, the complexity of topology search space is dependent on it. When we use different orders: $o_1$ and $o_2$, the graphlizer $f$ will map $G$ to different transformed graphs $f_{o_1}(G)$ and $f_{o_2}(G)$, where $f_{o_1}(G)$ and $f_{o_2}(G)$ may contain different numbers of subgraphs. Hence, the numbers of connections between subgraphs to encode in the outer message passing are different.

In the studied op-amp circuits automation problem, we find that the order of subgraphs in basis $\mathbb{B}$ will not affect the transformation from circuits $G$ to transformed graphs $f(G)$ (though it is not true for the general problems) due to the property of op-amp circuits. For instance, when we reverse the order of C and R (i.e from $o(R) > o(C)$ to $o(C) > o(R)$), the order of subgraphs 'C parallel gm' and 'R parallel gm' will also be reversed (i.e from $o('R\ parallel\ gm') > o('C\ parallel\ gm')$ to $o('C\ parallel\ gm') > o('R\ parallel\ gm')$). However, when the reversed subgraph order affects the transformation from $G$ to $f(G)$, we find that the gm, C, and R should be connected in parallel, which means that they formulate another subgraph 'gm parallel R parallel C' of larger size in the basis $\mathbb{B}$, which means that the graphlizer $f$ should choose subgraph 'gm parallel R parallel C' instead of decomposing it to smaller subgraphs. We also empirically evaluate the effect of domain specific total ordering of nodes on our dataset, and we find 0 of 10000 circuits $G$ have different transformed graphs $f(G)$ when we reverse the order of C and R.

## B    EXPRESSIVE ABILITY OF TWO-LEVEL GNN WITH A SUBGRAPH BSIS

Though various GNN models have been proposed, (Atwood & Towsley, 2016; Verma & Zhang, 2018; Niepert et al., 2016), most follow the message passing scheme (Gilmer et al., 2017). Message passing GNNs (MPGNNs) implicitly leverage the inductive bias in the graph topology by iteratively passing messages between each node and its neighbors to extract a node embedding that encodes the local substructure. The message passing scheme shows impressive graph representation learning ability to extract highly expressive features representing the local structure information in a graph. MPGNNs show a close relationship with the WL test, and (Xu et al., 2018) has proven that they are at

most as powerful as the WL test for distinguishing graph structures, which limits the expressive power of message passing GNNs. To improve the expressiveness of message passing GNNs, a plug-and-play framework, NGNN (Zhang & Li, 2021), is proposed to learn substructures where standard message passing scheme fails to discriminate. NGNN uses a two-level GNN framework that uses an inner GNN to learn the representation of extracted subgraphs around each node to embed the substructures, instead of rooted subtrees, thereby efficiently distinguishing almost all d-regular graphs.

**Corollary B.1** *By selecting proper subgraph basis $\mathbb{B}$ and order function $o$, the expressiveness of the two-level GNN framework with subgraph basis $\mathbb{B}$ can be: 1) strictly more powerful than message passing GNNs; 2) as powerful as nested GNN in distinguishing n-sized r-regular graphs.*

When the subgraph basis $\mathbb{B}$ contains all subgraphs of depth 1, the two-level GNN with $\mathbb{B}$ is equivalent to a general message passing framework. If we add rooted subgraphs with a height larger than 1, message passing in inner GNNs will be applied to added rooted subgraphs to extract additional distance features (Li et al., 2020) to provide further distinction ability. On the other hand, when the subgraph basis contains all height-$k$ rooted subgraphs and the order function $o$ defines a partial order dependent on the root node $v$, then the two-level GNN with subgraph basis $\mathbb{B}$ is essentially NGNN framework.

The Corollary indicates that we can manipulate the subgraph basis to balance the trade-off between the complexity and the expressiveness of the two-level GNN framework. We leave the problem in the future to incorporate the subgraph selection progress into the end-to-end learning framework for the general graph learning tasks. In this work, we focus on the circuit (DAG) optimization problem, where the ordered subgraph basis is known, and we show that it can be used to reduce the (topology) search space size of the optimization problem.

## C  PROOF OF THEOREM 3.2

It's straightforward that the $f$ can convert input DAG $G$ to some transformed DAGs $G'_1, ... G'_m$ if basis $\mathbb{B}$ contains every of size 1. Then we can identify the unique $f$ by following two rules:

(1) Then the output of $f$ should minimize the number of nodes (subgraphs used in the basis $\mathbb{B}$) in the transformed graph $G'$. That is. $f(G) = G' = \arg\min_{i=1,2,..m} |G'_i|$.

(2) If there exists two transformed subgraphs $G'_i$ and $G'_j$ such that $|G'_i| = |G'_j| = \arg\min_{l=1,2,..m} |G'_l| = K$, then both $G'_i$ and $G'_j$ are represented as $K$ subgraphs in basis $\mathbb{B}$. $g^i_1, .. g^i_K$. We denote these $K$ subgraphs as $G'_{i,1}, G'_{i,2},..., G'_{i,K}$ and $G'_{j,1}, G'_{j,2},..., G'_{j,K}$, then we can sort $o(G'_{i,1}), o(G'_{i,2}),..., o(G'_{i,K})$ to provide $G'_i$ a ordered tuple $t_i$ of size $K$, and sort $o(G'_{j,1}), o(G'_{j,2}),..., o(G'_{j,K})$ to provide $G'_j$ a ordered tuple $t_j$ of size $K$. Then the output of $f$ is given by:

$$f(G) = \begin{cases} G'_i, & t_i > t_j \\ G'_j, & t_j > t_i \end{cases}$$

where $>$ denotes the lexicographical sorting operation.

The above rules ensure that the graphitized $f$ can injectively select subgraphs in the basis $\mathbb{B}$ to formulate the transformed DAG $G'$. Then, since each subgraph in $\mathbb{B}$ only has a single node one node to be the head (tail) of a directed edge whose tail (head) is outside the subgraph, the connections between subgraphs are uniquely dependent on the input DAG $G$. Furthermore, as connections between nodes within each subgraph in $\mathbb{B}$ are fixed, the order of nodes in input $DAG$ is uniformly encoded in the transformed DAG $G'$.

## D  EXPERIMENT DETAILS AND ADDITIONAL RESULTS

**Model Configurations**    In the experiment, message passing GNNs (i.e. GCN and GIN) contain 3 graph convolution layers and use mean pooling to read out graph-level representation. NGNN uses a rooted subgraph height $h = 3$ and takes GIN with $l = 3$ layers as the base GNN. In Transformer-based models (i.e., Graphormer and PACE), the number of Transformer encoder layers is 5 and it

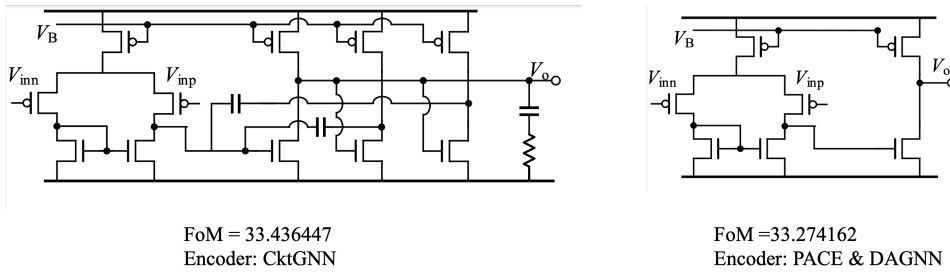



FoM = 33.436447
Encoder: CktGNN

FoM =33.274162
Encoder: PACE & DAGNN



Figure 8: Exemplary optimal circuits detected by Bayesian Optimization.

plays multi-head attention with 6 heads. DAGNN uses two-layer directed graph convolutions. We implement experiments on 12G TeslaP100 and GeForce GTX 1080 Ti, and 5 independent trials are implemented when the mean and standard deviation are reported.

**Bayesian Optimization Results**    In the experiment, we take the Bayesian optimization Approach and optimize the FoM score over the (latent) vectorial space generated by the DAG encoder. Figure 8 illustrates the results. Optimization on the latent space by CktGNN yields a circuit with the largest FoM score than competitive DAG encoders (or GNN for undirected graphs).

# E    MORE DETAILS ABOUT TRAINING AND EVALUATION

**Training Methodology** The proposed CktGNN can be trained in a supervised fashion, or in the VAE framework. When trained with supervised learning, the loss function is formulated with the target value and the output representation of CktGNN. On the other hand, CktGNN can also be trained in a VAE framework such as D-VAE, DAGNN, and PACE. In the VAE architecture, we take CktGNN as the DAG encoder. Two fully connected (FC) layers take the output representation of CktGNN as inputs to predict the mean and variance of the approximated posterior distribution in the evidence lower bound (ELBO) (Kingma & Welling, 2013). The decoder is formulated as the decoder of D-VAE to generate transformed DAGs $G^{'}$ from the latent space. That is, the decoder uses the same undirected message passing as CktGNN to learn intermediate node and graph states $z_{v'}$. For the $i^{th}$ generated node $v_i^{'}$ in $G^{'}$, MLPs are used to predict the subgraph type in the basis $\mathbb{B}$ and the existence of directed edges $(v_j^{'}, v_i^{'})$ for $v_j^{'} \in G^{'}$, $j = 1, 2, ..., i - 1$. In addition, given the predicted subgraph type, the topology within the subgraph is known. Therefore, the decoder only needs to use MLPs to predict the node features within the subgraph based on the current graph state $z_{v'}$.

In the training methodology, we resort to a VAE architecture, and the VAE loss is formulated as reconstruction loss + $\alpha$ KL divergence, where $\alpha$ is set to be $0.005$ as prior works (i.e. Grammer variational autoencoder Kusner et al. (2017), D-VAE, PACE). In the training process, we perform mini-batch stochastic gradient descent with a batch size of $64$. We train models for 200 epochs. The initial learning rate is set as 1E-4, and we use a schedule to modulate the changes of the learning rate over time such that the learning rate will shrink by a factor of $0.1$ if the training loss is not decreased for 20 epochs.

When evaluating the circuit design ability in section 5.4, we use two metrics: Valid circuits and Novel circuits, to measure the circuit design (generation) effectiveness. To calculate these values, we randomly sample 1000 random points in the latent vectorial space from a (prior) standard normal distribution, and decode each point for 10 times. Then we compute the proportion of valid circuits and novel circuits not seen in the training sets. A generated circuit is valid if it satisfies following three criteria: (1) It has only one input node and one output node; (2) The generated circuit (directed graph) is a DAG (i.e. there is no cycle); (3) In the main path, there is no R node nor C node. The last criterion is consistent with the design rules of operational amplifiers (Op-Amps), and more details are available in the Appendix A.

# F  MORE DETAILS ABOUT DECODER

Basically, the decoder is also based on the CktGNN framework where the GRU-based directed message passing (introduced in D-VAE) is used as the outer GNN. Given a point $z$ in the latent vectorial space, an MLP is used to map $z$ to $h_0$ as the initial hidden state. Next, the decoder constructs the circuits subgraph by subgraph, where subgraphs are from the known subgraph basis $\mathbb{B}$. Specifically, for the $i$th subgraph $v_i^{\cdot}$, we use an MLP $f_{subg}(h_{v_{i-1}^{\cdot}})$ to compute the subgraph type distribution, and sample the subgraph type $v_i^{\cdot}$. Then we can update the hidden state representation $h_{v_{i-1}^{\cdot}}$ using the CktGNN framework based on current circuit (DAG). Given the predicted subgraph type, the topology within the subgraph is known, thus the decoder only needs to use MLPs to predict the node features within the subgraph based on $h_{v_{i-1}^{\cdot}}$. In the end, for previous subgraph $v_j^{\cdot}$ such that $j < i$, we use an MLP $f_{edge}(h_{v_j^{\cdot}}, h_{v_i^{\cdot}})$ to predict the probability of connection between $v_j^{\cdot}$ and $v_i^{\cdot}$. These steps are repeated until the generated subgraph type is an output type.

