# OpenReview forum: "CktGNN:  Circuit Graph Neural Network for Electronic Design Automation"
_ICLR.cc/2023/Conference — ICLR 2023 poster_

### Official Review · Reviewer_uYkn · 2022-10-21

**Confidence:** 3
**Correctness:** 3
**Technical Novelty And Significance:** 2
**Empirical Novelty And Significance:** 4
**Recommendation:** 6

**Clarity, Quality, Novelty And Reproducibility:**

Novelty:
CktGNN almost adopts the same structure as nested GNN, while only altering the outer GNN to DAG learning. Also, in the section discussing the method, a significant part of the content is about nested GNN, 1-WL algorithm, and VAE, making me feel that the novel part is limited.

Clarity:
Most of the paper is quite clear. The description and the purpose of Theorem 3.2 might require more revision.

Reproducibility:
The code is available in the supplementary material.

**Strength And Weaknesses:**

Strength:
1. The benchmark dataset has an empirical contribution to the field.
2. CktGNN is empirically proven to achieve SOTA performance.

Weakness:
1. The discussion on the decoder (circuit design/generation) is too little.
2. The method is an incremental development upon the nested GNN.
3. The figures are hard to read due to the font size.
4. The ability to design circuits automatically is not evaluated adequately in the experiments.

**Summary Of The Paper:**

This paper designs a circuit GNN (CktGNN) to facilitate the learning of circuit graphs and electronic design automation. CktGNN is a generation of the nested GNN structure to DAGs. The paper also presents a circuit benchmark dataset OCB with open-source code for automated circuit design methods evaluation.

**Summary Of The Review:**

The benchmark dataset provided has an unneglectable contribution to the field. However, the method itself doesn't offer much novelty. It's barely an application of the nested GNN.

---

> ### Author Response · Authors · 2022-11-12
> **Author Response 1/2**
>
> Thanks for the helpful comments. We address your concerns below.
>
> $\textbf{Q1:}$ The discussion on the decoder (circuit design/generation) is too little.
>
> $\textbf{A1:}$ Here we provide more details about how the decoder of the automation framework generates circuits from the latent space, and we have added the discussion in Appendix F in the revised paper.
>
> Basically, the decoder is also based on the CktGNN framework where the GRU-based directed message passing (introduced in D-VAE) is used as the outer GNN. Given a point $z$ in the latent vectorial space, an MLP is used to map $z$ to $h_{0}$ as the initial hidden state. Next, the decoder constructs the circuits subgraph by subgraph, where subgraphs are from the known subgraph basis $\mathbb{B}$.
> Specifically, for the $i$th subgraph $v_{i}^{'}$,  we use an MLP $f_{subg} (h_{v_{i-1}^{'}})$ to compute the subgraph type distribution, and sample the subgraph type $v_{i}^{'}$. Then we can update the hidden state representation $h_{v_{i-1}^{'}}$ using the CktGNN framework based on the current circuit (DAG). Given the predicted subgraph type, the topology within the subgraph is known, thus the decoder only needs to use MLPs to predict the node features within the subgraph based on $h_{v_{i-1}^{'}}$. In the end, for previous subgraph $v_{j}^{'}$ such that $j < i$, we use an MLP $f_{edge}(h_{v_{j}^{'}}, h_{v_{i}^{'}})$ to predict the probability of connection between $v_{j}^{'}$ and $v_{i}^{'}$. These steps are repeated until the generated subgraph type is an output type.
>
> $\textbf{Q2:}$ The figures are hard to read due to the font size.
>
> $\textbf{A2:}$ We have adjusted the figures in the revised paper.
>
>
> $\textbf{Q3:}$ The ability to design circuits automatically is not evaluated adequately in the experiments.
>
> $\textbf{A3:}$  Basically, the ability to automatically design (generate) circuits can be evaluated from the following two aspects: (1) the quality of generated circuits of the automation framework and (2) the circuit generation efficiency.
>
> In the paper (please see Table 2), we use three metrics to evaluate the ability of our automation framework to automatically design circuits: (1) the proportion of valid circuits in the generation process (Valid circuits); (2) the proportion of novel circuits (Novel circuits); (3) The Figure of Merit (FoM) value of the best circuit detected by the Bayesian optimization algorithm (BO(FoM)). The first and second metrics evaluate how often our automation algorithm can generate valid circuits and the proportion of valid circuits that are never seen in the training sets. Thus, higher values indicate a better exploratory $\textbf{efficiency}$, which provides better applicability in the real-world circuit design process. We can take the average value of Valid circuits and Novel circuits to compare the applicability, and we find CktGNN achieves the highest average value of these two metrics. On the other hand, the third metric directly measures the $\textbf{quality}$ of output circuits of our automation framework, and a higher value indicates that the automation algorithm can detect better circuits. We also provide the designed circuit in Appendix D, and it shows that CktGNN-based automation framework can find interesting unconventional circuit topology that meets the design targets, indicating that the proposed automation framework can help to discover innovative circuit topologies that have not been studied.
>
> Second, we also evaluate the design automation ability of our method by using design efficiency, i.e., the time taken to achieve an analog circuit design in both topology and device parameters. Table.1 shows the comparison between our proposed method and the manual design method. In the manual design loop, a human expert (a senior post-doc researcher) often spends several hours searching for a proper circuit topology, based on which the expert also uses tens and even hundreds of fine iterations to search optimal device parameters to realize the desired circuit specifications. The whole design process usually takes ~10 hours. In contrast, once trained, our method often generates a feasible Op-Amp with a proper topology and optimal device parameters within one minute. Despite the training cost, our proposed model can averagely achieve a much higher design efficiency at the order of 1000x compared to human experts by designing 10 Op-Amps.
>
>
> Table 1. Comparison of design efficiency between our method and the manual design by evaluating the design of 10 Op-Amps.
>
> |Methods | Average design time |Average design efficiency |
> | :------:| :------: | :------: |
> |Our method|<1 min| ~1000X|
> |Manual design|~10 hours|1X|

---

> ### Author Response · Authors · 2022-11-12
> **Author Response 2/2**
>
> $\textbf{Q4:}$ Novelty concern: The benchmark dataset provided has an unneglectable contribution to the field. However, the method itself doesn't offer much novelty. It's barely an application of the nested GNN. The method is an incremental development upon the nested GNN.
>
> $\textbf{A4:}$ We appreciate the reviewer’s comment and would like to discuss our innovations in this paper here.
> The novelty of our method is at least three-fold spanning application, dataset, and algorithm rather than limited by the incremental development upon the nested GNN.
>
> First of all, our method is a non-trivial application of machine learning. We apply it to tackle the long-standing challenge of analog circuit design automation in the integrated circuit field. Particularly, we target the very challenging problem of simultaneously optimizing circuit topologies and device parameters which have been limitedly addressed so far. The conventional design of analog circuits follows a manual flow with time-consuming and cost-expensive labor efforts. Automated design of analog circuits has thus been long sought-after and can profoundly benefit the entire electronic ecosystem. Our method shows the promise and power of representation learning to automate the whole schematic-level design of analog circuits. With the growing research efforts (i.e. Mirhoseini et al., 2021 and Khailany et al., 2020) in this field, we believe that learning-assisted electronic design automation will be as bright as other applied machine learning fields such as robotics, computer vision, drug discovery, etc.
>
> Second, as acknowledged by the reviewer, we released the first OCB dataset for analog circuit topology and device parameter optimization. Conventionally, analog circuits are proprietary and there is a lack of public benchmarks to evaluate various electronic design automation methods. Our dataset ameliorates the issue and provides a standard benchmark for reproducible research and empirical evaluations.
>
> Last but not the least, our method also has contributions from the proposed two-level DAG encoding framework (CktGNN). Although CktGNN extends the two-level GNNs framework (i.e. NGNN) to the circuit (DAG) encoding problem based on an ordered subgraph basis, it is not a straightforward combination of these existing methods, which is also confirmed by reviewer ESXP: “Novelty might be argued against as it is an extension of an existing work, but I think the method found is non-obvious and properly analysed.” We explain the reasons below.
>
> (1) NGNN (two-level GNN framework) is originally developed to increase the expressiveness of GNNs for undirected graphs, and it fails to encode dependency between nodes in DAGs. The experimental results (please see Table 1 in the paper) also show that NGNN is less competitive than directed GNNs (i.e. D-VAE, DAGNN, etc) in the circuit (DAG) encoding problem. To alleviate this problem, CktGNN proposes to encode the dependency between \textbf{non-overlapping} subgraphs in DAGs through the outer GNN in CktGNN, and this intuition is consistent with the fact that circuits can be constructed with some basic subgraph blocks (which is also pointed out by reviewer F1jp: “The two-level GNN is natural for the analog circuit problem as the circuit needs to be constructed with some basic blocks instead of random circuit components.”) (2) In addition, compared with NGNN, CktGNN doesn’t extract a subgraph around each node to implement the inner GNN. Instead, as the answer to Q3 of reviewer ESXP shows, CktGNN only needs to perform inner GNNs for decomposed non-overlapping subgraphs in input circuits (DAGs) to provide expressive embeddings. Hence, CktGNN can significantly reduce the number of subgraphs to perform inner GNNs when most subgraphs in the basis have relatively large sizes. Many circuit design problems have large designed (subgraph) blocks. Thus, CktGNN is more suitable for these practical circuit design problems. (3) In the end, CktGNN also wisely uses the two-level GNN framework to solve the notorious circuit encoding problem with complex topologies and node features: (3.1) The inner GNNs in CktGNN enable more information aggregation iterations than general DAG encoder (i.e. D-VAE, DAGNN) to increase the ability to capture the contextualized information in circuits; (3.2) The outer GNN in CktGNN only needs to encode the dependencies between subgraphs (while other DAG encoders need to encode dependencies between each node pair), which successfully shrinks the topology search space.

---

> ### Comment · Reviewer_uYkn · 2022-11-15
> **Raise recommendation from 5 to 6**
>
> After reviewing the revisions in the newest version, I can happily raise the recommendation from 5 to 6: marginally above the acceptance threshold.

---

> > ### Author Response · Authors · 2022-11-19
> > **A Friendly Reminder**
> >
> > We appreciate your time and efforts in reviewing all these comments and the revision, and we are happy that they can address some of your concerns. We would like to friendly remind you that the recommendation score in the official review is not changed yet, and we sincerely hope that you could officially edit it at your convenience.

---

> > > ### Comment · Reviewer_uYkn · 2022-11-25
> > > **Thanks for the reminder**
> > >
> > > Raised!

---

### Official Review · Reviewer_ESXP · 2022-10-24

**Confidence:** 4
**Correctness:** 3
**Technical Novelty And Significance:** 3
**Empirical Novelty And Significance:** 3
**Recommendation:** 8

**Clarity, Quality, Novelty And Reproducibility:**

The paper is clearly written (all important concepts are conveyed clearly), the evaluation needs more details (see weaknesses) and the overall quality is acceptable. Novelty might be argued against as it is an extension of an existing work,but I think the method found is non-obvious and properly analysed, with the dataset being a cherry on top.


there is also a typo in sectoin 3.3 "trasnformed"

**Strength And Weaknesses:**

Strengths:

- the paper is well written and easy to understand
- the architecture achieves decent gains against most competing models
- the architecture should be useful for other DAG centric tasks as well

Weaknesses

- an author claim that this is the "first open source dataset for analog eda" should be made more nuanced, given the existence of e.g. https://arxiv.org/abs/2208.01040 and https://arxiv.org/abs/2203.15913 and their accompanying datasets
- the method relies on domain specific total ordering which needs to be handcrafted, similar to the node ordering methods used by GraphRNN and GRAN. The relation and impact of these should maybe be discussed, and if there is time, and ablation over the sensitivity towards ordering choice (e.g.  setting o(C)>o(R))  would strengthen the paper
- either I missed them or the experimental details (optimiser choice, number of seeds etc) are missing and should be added, although they are in the supplementary code

**Summary Of The Paper:**

The paper presents a modification of the NGNN framework of Zhang & Li which uses the embeddings of subgraphs around each node as node embeddings for an outer global message passing loop to DAGs (by defining a total ordering of possible subgraphs and  selecting the largest for each node) in order to apply the architecture to EDA tasks. The expressiveness of the architecture is analysed showing that the method is more expressive than standard MPGNNs and a new dataset is proposed on which the method and related work are evaluated.

Contributions:

1. domain knowledge dependent subgraph extraction for DAGs to extend NGNN
2. expressivty analysis
3. dataset + generation codes
4. empirical evaluation

**Summary Of The Review:**

Overall this is a fine paper. I would not call it ground breaking, but the dataset + code generation takes considerate amount of work to accomplish and will be valuable for the community, the method improves on prior art (not a lot but noticably) and the modification is well justified and analysed.

---

> ### Author Response · Authors · 2022-11-12
> **Author Response 1/2**
>
> We appreciate the reviewer’s constructive feedbacks and positive review.
>
> $\textbf{Q1:}$ The claim that  "first open source dataset for analog EDA" should be made more nuanced.
>
> $\textbf{A1:}$ Thank you very much for sharing us with these references and helping us to improve our claim here.
>
> We carefully digest these references and find that https://arxiv.org/abs/2208.01040 is the first open-sourced dataset for digital circuits, i.e., RISC-V. https://arxiv.org/abs/2203.15913 is the first open-sourced dataset for circuit voltage prediction. Note that this dataset is built upon collecting data points from neural network prediction rather than circuit-level simulations. Our dataset is the first open-source dataset to optimize circuit topologies and device parameters, which contains detailed circuit graphs, circuit netlists, and many DC and AC specifications that are obtained from circuit-level simulations.
>
> To clarify our claim, we revise the texts correspondingly which are highlighted in blue in the introduction. We also attached them below for the reference of the reviewer.
>
> "To ameliorate the issue, we introduce Open Circuit Benchmark (OCB), the first open graph dataset for optimizing both analog circuit topologies and device parameters, which is a good supplement to the growing open-source research in the electronic design automation community for IC (Chai et al., 2022; Hakhamaneshi et al., 2022) ."
>
> $\textbf{Q2:}$ Additional experimental details  (optimizer choice, number of seeds, etc.)
>
> $\textbf{A2:}$ Here we provide more experimental details, and we include them in Appendix E in the revised paper.
>
> In the training methodology, we resort to a VAE architecture, and the VAE loss is formulated as reconstruction loss + $\alpha$ KL divergence,  where $\alpha$ is set to be 0.005 as prior works (i.e. Grammer variational autoencoder [1], D-VAE, PACE). In the training process, we perform mini-batch stochastic gradient descent with a batch size of 64. We train models for 200 epochs. The initial learning rate is set as 1E-4, and we use a schedule to modulate the changes of the learning rate over time such that the learning rate will shrink by a factor of $0.1$ if the training loss is not decreased for 20 epochs.
>
> When evaluating the circuit design ability in section 5.4, we use two metrics: Valid circuits and Novel circuits, to measure the circuit design (generation) effectiveness. To calculate these values, we randomly sample 1000 random points in the latent vectorial space from a (prior) standard normal distribution and decode each point for 10 times. Then we compute the proportion of valid circuits and novel circuits not seen in the training sets. A generated circuit is valid if it satisfies the following three criteria: (1) It has only one input node and one output node; (2) The generated circuit (directed graph) is a DAG (i.e. there is no cycle); (3) In the main path, there is no R node nor C node. The last criterion is consistent with the design rules of operational amplifiers (Op-Amps), and more details are available in Appendix A.
>
> [1] Matt J Kusner, Brooks Paige, and José Miguel Hernández-Lobato. Grammar variational autoencoder. In International Conference on Machine Learning, pages 1945–1954, 2017.

---

> ### Author Response · Authors · 2022-11-12
> **Author Response 2/2**
>
> $\textbf{Q3:}$ The impact of domain specific total ordering of nodes, and the relations to other node-order-dependent models like GraphRNN and GRAN.
>
> $\textbf{A3:}$ Thanks for the insightful question and suggestion. Indeed, various node orderings have been adopted in encoders for the DAG encoding problem, and node ordering is especially important for auto-regressive models. For instance, S-VAE, D-VAE and DAGNN use topological ordering, PACE and GRAN [1]  use canonical ordering, and graphRNN [2] uses  BFS ordering. The key of developing powerful DAG encoders is to make sure that the node-order dependent encoding process can injectively map non-isomorphic DAGs (or different computations of DAGs) to different embeddings in the vectorial space, and experimental results in prior works (PACE, D-VAE, DAGNN) have confirmed the claim on general DAG encoding tasks, such as NAS (neural architecture search) and Bayesian network structure learning. In summary, PACE can injectively encode the structure of DAGs by linearizing DAGs as sequences according to the canonical ordering, while D-VAE and DAGNN can injectively encode computation of DAGs following an RNN-based directed message passing. Compared with these methods, S-VAE and GraphRNN-like encoder (GraphRNN is originally designed as a pure generative model, yet it can be extended to a BFS-order based DAG encoder following the idea of S-VAE, and details are available in the ablation study section in the paper of PACE) fail to injectively encode structure or computation of DAGs as both topological ordering and BFS ordering are not unique. As a result, PACE, D-VAE, and DAGNN significantly outperform S-VAE and GraphRNN.
>
> Following the above analysis, we find that the expressive ability of CktGNN is independent of domain specific total ordering of nodes. Basically, the total ordering of nodes only determines the order of subgraphs (i.e. $o$) in the subgraph basis $\mathbb{B}$ (please see the last paragraph in Appendix A). When basis $\mathbb{B}$ satisfies the conditions in Theorem 3.2, there exists an injective graphlizer $f$, and $f$ can be formulated as Appendix C suggests. Hence, for $\textbf{any given order}$ $o$, the input graph $G$ is injectively mapped to $f(G)$, based on which CktGNN performs the two-level GNNs. Then, if both the inner GNNs and outer directed message passing are injective, CktGNN can injectively map $f(G)$ to embeddings in the vectorial space, as injective functions follow the transitive property. Hence, for any order $o$ on subgraph basis $\mathbb{B}$, the overall encoding process can injectively map $G$ to embeddings.
>
> Although the expressive ability is not affected by the node ordering, the complexity of the topology search space is dependent on it. When we use different orders: $o_{1}$ and $o_{2}$, the graphlizer $f$ will map $G$ to different transformed graphs $f_{o_{1}}(G)$ and $f_{o_{2}}(G)$, where $f_{o_{1}}(G)$ and $f_{o_{2}}(G)$ may contain different numbers of subgraphs. Hence, the numbers of connections between subgraphs to encode in the outer message passing are different.
>
> In the studied op-amp circuits automation problem, we find that the order of subgraphs in basis $\mathbb{B}$ will not affect the transformation from circuits $G$ to transformed graphs $f(G)$ (though it is not true for the general problems) due to the property of op-amp circuits. For instance, when we reverse the order of C and R (i.e from $o(R) > o(C)$ to $o(C) > o(R)$), the order of subgraphs 'C parallel gm' and 'R parallel gm' (Please see Figure 7 in Appendix A) will also be reversed (i.e from $o('R\ parallel\ gm') > o('C\ parallel\ gm')$ to $o('C\ parallel\ gm') > o('R\ parallel\ gm')$). However, when the reversed subgraph order affects the transformation from $G$ to $f(G)$, we find that the gm, C, and R should be connected in parallel, which means that they formulate another subgraph 'gm parallel R parallel C' of larger size in the basis $\mathbb{B}$, which means that the graphlizer $f$ should choose subgraph ‘gm parallel R parallel C’ instead of decomposing it to smaller subgraphs. We also empirically evaluate the effect of domain specific total ordering of nodes on our dataset, and we find 0 of 10000 circuits $G$ have different transformed graphs $f(G)$ when we reverse the order of C and R. Above analysis is added at the end of Appendix A in the revised paper.
>
> [1] Liao, Renjie, Yujia Li, Yang Song, Shenlong Wang, Will Hamilton, David K. Duvenaud, Raquel Urtasun, and Richard Zemel. "Efficient graph generation with graph recurrent attention networks." Advances in neural information processing systems 32 (2019).
>
> [2] Jiaxuan You, Rex Ying, Xiang Ren, William Hamilton, and Jure Leskovec. Graphrnn: Generating realistic graphs with deep auto-regressive models. In International Conference on Machine Learning, pages 5694–5703, 2018.

---

### Official Review · Reviewer_F1jP · 2022-10-25

**Confidence:** 5
**Correctness:** 4
**Technical Novelty And Significance:** 3
**Empirical Novelty And Significance:** 3
**Recommendation:** 6

**Clarity, Quality, Novelty And Reproducibility:**

Clarity and quality is good, except that the paper should emphasize more on its representation's ability to be applied to different tasks.
novelty is good.

**Details Of Ethics Concerns:**

No ethics concerns

**Strength And Weaknesses:**

Strength:
The two-level GNN is natural for the analog circuit problem as the circuit needs to be constructed with some basic blocks instead of random circuit components.
The proposed benchmark is important for future study in this field

Weakness:
The OCB dataset only contains the operational amplifiers, though Opamp is a widely used units, it would be good to include other important circuit such as LDO, comparators, etc.
It is unclear that whether the two-level circuit representation can be used to other circuit since the subgraph basis may not be generally applied to all kinds of circuit.

Comments:
How can this method benefit the layout design of analog circuits? is that possible to design a layout for each subgraph basis?



**Summary Of The Paper:**

The paper studies the problem of representation learning of the analog circuit. It propose a two-level GNN model, in which the outer graph is composed with multiple non overlapping subgraphs. The representation can be used to perform different tasks. Then it also present a dataset called Open Circuit Benchmark which contains 10K OpAmp to benefit future projects.

**Summary Of The Review:**

A new representation learning for analog circuit which leverages human knowledge to construct the high level graph with low level subgraphs. A circuit dataset is released containing 10K Opamp circuits.

Thanks for the response. I would like to increase the score to 8 and i would encourage the authors to add more circuit to the dataset.

---

> ### Author Response · Authors · 2022-11-12
> **Author Response**
>
> We appreciate the reviewer’s positive feedback and thoughtful comments.
>
> $\textbf{Q1:}$ The OCB dataset only contains the operational amplifiers, though Opamp is a widely used units, it would be good to include other important circuits such as LDO, comparators, etc. It is unclear whether the two-level circuit representation can be used to other circuits since the subgraph basis may not be generally applied to all kinds of circuits.
>
> $\textbf{A1:}$ The reviewer raises a very important point regarding the generalization ability of our proposed method. We would like to carefully discuss it here.
>
> Our method is generalizable to design various analog circuits with minimal modifications. Op-Amps are the most essential building blocks in many modern analog circuits. Typically, LDOs and comparators are built upon Op-Amps with several external feedback/feedforward paths. With this key insight in mind, the two-level circuit representation can be seamlessly applied to design LDOs and comparators by including the essential subgraph bases of the external feedback/feedforward paths. Due to the time-consuming circuit-level simulations but the limited time budget, we may not update our OCB dataset timely with sufficient LDOs and comparators. However, we are striving to provide as many as possible examples in the dataset before the end of the discussion stage. Please stay tuned.
>
> Despite the promise of the proposed method, we agree with the reviewer that the subgraph basis for Op-Amps may not be applicable to all kinds of analog circuits given their huge design space and abundant types. Exploring a general method to design arbitrary analog circuits is of great significance and may be beyond the scope of this paper. We hope to further investigate it in our future work.
>
> $\textbf{Q2:}$ How can this method benefit the layout design of analog circuits? Is that possible to design a layout for each subgraph basis?
>
> $\textbf{A2:}$  We very much appreciate the reviewer's insightful and farsighted view and would like to discuss it in detail here.
>
> The layout design and the schematic design are often independent but equally important problems in analog circuit electronic design automation. Physical layout design has many unique challenges distinct from the schematic design. Therefore, we envision that leveraging the proposed CktGNN framework for the physical design would demand significant further developments. However, the reviewer’s comment is quite inspiring and we believe that designing a layout for each subgraph basis can be a promising way to extend our CktGNN framework to automate the physical layout design of analog circuits, which we intend to investigate in the future.

---

> ### Author Response · Authors · 2022-11-19
> **Follow-up Response to $\textbf{Q1:}$**
>
> We appreciate the reviewer's comment again. However, we sincerely apologize that we may not update a reasonable size of LDOs and comparators in our benchmark at this stage due to several practical constraints. First, we tried to search for tens of LDOs [1-5] and comparators from previous literature but found that they were designed with diverse technologies, such as bipolar and out-of-date CMOS technology or there are no clear device parameters for easy replication. Porting those designs into our server with an advanced CMOS technology node is non-trivial and actually very time-consuming as we need to manually tune the circuits to ensure they work correctly. Therefore, although we are struggling to design these circuits by ourselves, only a very limited size of data points are collected. Second, due to the prior limitation, we found that the CktGNN models (so as competitive baselines) corresponding to such circuits cannot be trained well with extremely limited training data. It is hard to demonstrate the performance when applying our method to design these circuits.
>
> For these factors, we are far away to provide an updated dataset. Nonetheless, we are continuously augmenting our data points and wish to update a reasonable size of datasets for such circuits when finally releasing our OCB.
>
> [1]  S. K. Lau, P. K. T. Mok and K. N. Leung, "A Low-Dropout Regulator for SoC With Q-Reduction," in IEEE Journal of Solid-State Circuits, vol. 42, no. 3, pp. 658-664, March 2007, doi: 10.1109/JSSC.2006.891496.
>
> [2] X. L. Tan, S. S. Chong, P. K. Chan and U. Dasgupta, "A LDO Regulator With Weighted Current Feedback Technique for 0.47 nF–10 nF Capacitive Load," in IEEE Journal of Solid-State Circuits, vol. 49, no. 11, pp. 2658-2672, Nov. 2014, doi: 10.1109/JSSC.2014.2346762.
>
> [3] Q. -H. Duong et al., "Multiple-Loop Design Technique for High-Performance Low-Dropout Regulator," in IEEE Journal of Solid-State Circuits, vol. 52, no. 10, pp. 2533-2549, Oct. 2017, doi: 10.1109/JSSC.2017.2717922.
>
> [4] A. P. Patel and G. A. Rincón-Mora, "High Power-Supply-Rejection (PSR) Current-Mode Low-Dropout (LDO) Regulator," in IEEE Transactions on Circuits and Systems II: Express Briefs, vol. 57, no. 11, pp. 868-873, Nov. 2010, doi: 10.1109/TCSII.2010.2068110.
>
> [5] B. Razavi, "The Design of An LDO Regulator [The Analog Mind]," in IEEE Solid-State Circuits Magazine, vol. 14, no. 2, pp. 7-17, Spring 2022, doi: 10.1109/MSSC.2022.3167308.

---

### Official Review · Reviewer_nGNe · 2022-10-27

**Confidence:** 2
**Clarity, Quality, Novelty And Reproducibility:** Read section "Strength And Weaknesses"
**Correctness:** 3
**Technical Novelty And Significance:** 4
**Empirical Novelty And Significance:** 4
**Recommendation:** 6

**Strength And Weaknesses:**

Strength:
- Paper is well written and easy to follow
- The idea to simultaneously automate the circuit topology generation and device sizing is novel.

Weaknesses:

Admittedly, this paper does not fall into my area of expertise. So I cannot write a proper critical review. Though, I have several questions regarding the statements in the paper:

- In the paper it says: "previous work, inherently fail to optimize the circuit topology". This claim needs justification which I could not find in the paper.
- In the paper it says: "However, these methods often lead to the generation of many useless circuit topologies". Again, This claim is provided with no justification.
- Regarding the OCB dataset, who has designed the 10k operational amplifiers? Basically, the question is: where do they come from?
- In the paper it says: "We also find that CktGNN can achieve the highest novelty in the generation process". However, in Table 2, CktGNN  is not the one with the highest novelty score!
- There are some spelling errors in the text


**Summary Of The Paper:**

This work proposes a Graph Neural Network (CktGNN) to automate the design for both the device parameters (device sizing) and the circuit topology.
Furthermore, the paper also proposes an open source dataset, Open Circuit Benchmark (OCB), which consists of 10k operational amplifiers with detailed circuit specifications.

The proposed CktGNN, shows promising results in three different domain:
- the ability to predict the graph properties (i.e., Gain, BW, PM, FoM)
- Circuit encoding efficiency (training/inference time)
- new circuit generation


**Summary Of The Review:**

As mentioned before, this paper does not fall into my area of expertise. Hence, my decision is going to heavily rely on other reviewers' evaluation.

---

> ### Author Response · Authors · 2022-11-12
> **Author Response 1/2**
>
> We appreciate the positive feedback.
>
> $\textbf{Q1:}$ In the paper it says: "previous work, inherently fails to optimize the circuit topology". This claim needs justification which I could not find in the paper.
>
> $\textbf{A1:}$ We thank the reviewer’s comment to help us improve the claim here.
>
> Prior works (e.g., Wang et al., 2020; Liu et al., 2021; Cao et al., 2022a;b) are GNN-based ad-hoc approaches to tackling the device sizing problem, i.e., searching for optimal device parameters to fulfill desired circuit specifications with a given analog circuit topology. Under this context, when applying GNNs to the problem, they build GNN models based on the fixed circuit topology and encode device parameters as graph node features. During the entire learning/optimization process, only the node features (i.e., device parameters) are updated while the GNN structures (i.e., circuit topologies) remain unchanged. These works do not target the topology design and are thus unable to optimize circuit topologies.
>
> We revise the text correspondingly in the introduction to make the claim justified and also attach it below for the reviewer’s reference.
> “ While these methods promisingly outperform traditional heuristics (Liu et al., 2017) in node feature sizing (i.e., device sizing), they are not targeting the circuit topology optimization/generation, which, however, constitutes the most critical and challenging task in analog circuit design.”
>
> [1] Wang, H., Wang, K., Yang, J., Shen, L., Sun, N., Lee, H.S. and Han, S., 2020, July. GCN-RL circuit designer: Transferable transistor sizing with graph neural networks and reinforcement learning. In 2020 57th ACM/IEEE Design Automation Conference (DAC) (pp. 1-6). IEEE.
>
> [2] Liu, M., Turner, W.J., Kokai, G.F., Khailany, B., Pan, D.Z. and Ren, H., 2021, February. Parasitic-aware analog circuit sizing with graph neural networks and bayesian optimization. In 2021 Design, Automation & Test in Europe Conference & Exhibition (DATE) (pp. 1372-1377). IEEE.
>
> $\textbf{Q2:}$ In the paper it says: "However, these methods often lead to the generation of many useless circuit topologies". Again, This claim is provided with no justification.
>
> $\textbf{A2:}$ We thanks again for the reviewer’s comment to help us improve this claim.
>
> Prior efforts such as Genetic Algorithms (Das & Vemuri, 2007) leverage genetic operations, e.g., crossover, and mutation, to generate new circuit topologies. Particularly, these operations randomly combine existing sub-topologies from parental circuits or newly-generated sub-topologies from mutation to form off-spring circuits. Practical constraints from feasible circuit topologies are thus not incorporated into its generation process. Due to these factors, the generated topologies are often non-functional and ill-posed, making them unsuitable for practical analog circuits.
>
> To make the claim more justified, we revise the text correspondingly in section 2.2 and attach it below for the reviewer’s reference.
>
> "These works leverage genetic operations such as crossover and mutation to randomly generate circuit topologies and do not sufficiently incorporate practical constraints from feasible circuit topologies during the generation process. Therefore, most of those generated topologies are often non-functional and ill-posed."
>
> $\textbf{Q3: }$ Regarding the OCB dataset, who has designed the 10k operational amplifiers? Basically, the question is: where do they come from?
>
> $\textbf{A3:}$ The 10K operational amplifiers (Op-Amps) in the OCB dataset are carefully designed by the entire conversion-generation-simulation process as illustrated in part of Figure 2 and elaborated in Section 4 of the manuscript. Specifically, we first leverage the circuit-to-graph mapping to convert a general Op-Amp topology into a directed acyclic graph (the Graphlizer arrow in Figure 2). The details of this conversion process are provided in Appendix A. Based on this mapping strategy, various useful graphs corresponding to Op-Amp topologies can then be sampled and generated. A conversion-back process is subsequently used to transfer the generated Op-Amp topologies into circuit netlists which are finally simulated by circuit simulators to obtain the performance (i.e., circuit specifications). Therefore, 10K such sampled graphs with their performance metrics are compiled as our OCB dataset. In fact, with the generation code of this process, OCB can provide datasets of arbitrary size for learning. We use 10K samples due to the concern of circuit-level simulation time.
>
> To clarify the dataset generation process, we revise the texts in Section 4 correspondingly which are attached below for reviewer’s reference:
>
> “We leverage this conversion-generation-simulation loop to generate 10, 000 different Op-Amps with detailed circuit specifications.”

---

> ### Author Response · Authors · 2022-11-12
> **Author Response 2/2**
>
> $\textbf{Q4:}$ In the paper, it says: "We also find that CktGNN can achieve the highest novelty in the generation process". However, in Table 2, CktGNN is not the one with the highest novelty score!
>
> $\textbf{A4:}$ Thanks for pointing out this error, and we have fixed it in the revised paper as “We also find that CktGNN has the best circuit design (generation) ability.” Basically, in section 5.4 (and Table 2), the metric ‘Valid circuits’ measures the proportion of valid circuits generated by our automation framework, and the metric ‘Novel circuits’ measures the proportion of valid circuits that are unseen in the training dataset. Thus, the average value of these two metrics (i.e. ‘Valid circuits’ and ‘Novel circuits’) uniformly reflects the circuit design ability, and a higher value indicates that the circuit generation process is capable of providing more valid circuits to test, while discovering novel unseen circuits at the same time.
>
> $\textbf{Q5:}$ There are some spelling errors in the text
>
> $\textbf{A5:}$ We try our best to resolve the spelling errors in the text, such as
>
> 1. In Section 3.1, “the massage passing framework is given by:” We change the “massage” into “message”;
>
> 2. In Section 3.1, “the rules to extract subgraphs for inner GNNs to learn representations are as following” We change “following” into “follows”.
>
> 3. In Section 3.3,  “CktGNN will map the same input circuit (DAG $G$) to different trasnformed,” We change the “trasnformed” into “transformed”.
>
> 4. In Section 3.3, “if each subgraph $g_{v^{'}} \in \mathbb{B}$ has only one node to be the head (tail) of an directed edge whose tail (head) is outside the subgraph $g_{v^{'}}$.” We change the “an” here into “a”.
>
>  We would appreciate it a lot if the reviewer can also point it out in case we miss anyone.

---

### Author Response · Authors · 2022-11-12
**General Author Responses**

We’d like to thank all reviewers for their effort and constructive feedback in reviewing our paper. We are glad that our contributions have been well-received overall:

1. Quote from reviewer nGNe “The idea to simultaneously automate the circuit topology generation and device sizing is novel.”

2. Quote from Reviewer F1jp “The two-level GNN is natural for the analog circuit problem.”

3. Quote from reviewer F1jp “The proposed benchmark is important for future study in this field.”

4. Quote from reviewer ESXP “The architecture achieves decent gains against most competing models.”

5. Quote from reviewer ESXP “Novelty might be argued against as it is an extension of an existing work, but I think the method found is non-obvious and properly analysed, with the dataset being a cherry on top.”

6. Quote from reviewer uYkn “CktGNN is empirically proven to achieve SOTA performance.”

There seem to be a few concerns from reviewer uYkn based on the novelty. We kindly wish that our responses as well as the comments from other reviewers help to alleviate these issues. We fix typos and include additional clarifications in blue in the revised paper. In the following, we will address the reviewers’ comments individually.

---

### Decision · Program_Chairs · 2023-01-20

**Decision:**

Accept: poster

**Justification For Why Not Higher Score:**

The application area of analog circuit design is rather specialized, so this paper may be of interest to a smaller subset of the ICLR community.

**Justification For Why Not Lower Score:**

The reviewers are broadly supportive of this paper. This makes a contribution to a difficult and interesting application area.

**Metareview: Summary, Strengths And Weaknesses:**

This paper proposes a new benchmark for the problem of automatic design of analog circuits, and introduces a new type of graph neural network for this task, which combines a recent two-level GNN, but adapts it to directed graphs. Although this is a relatively small extension, it is a good fit for this challenging application, and works convincingly better than the base method.

**Note From Pc:**

if the above contains the word "oral" or "spotlight" please see: "oral" presentation means -> notable-top-5% and "spotlight" means -> notable-top-25%. As stated in our emails, we are disassociating presentation type from AC recommendations